# SELF-CONDITIONED EMBEDDING DIFFUSION FOR TEXT GENERATION

## ABSTRACT

Can continuous diffusion models bring the same performance breakthrough on natural language they did for image generation? To circumvent the discrete nature of text data, we can simply project tokens in a continuous space of embeddings, as is standard in language modeling. We propose Self-conditioned Embedding Diffusion (SED), a continuous diffusion mechanism that operates on token embeddings and allows to learn flexible and scalable diffusion models for both conditional and unconditional text generation. Through qualitative and quantitative evaluation, we show that our text diffusion models generate samples comparable with those produced by standard autoregressive language models — while being in theory more efficient on accelerator hardware at inference time. Our work paves the way for scaling up diffusion models for text, similarly to autoregressive models, and for improving performance with recent refinements to continuous diffusion.

## 1 INTRODUCTION

Continuous diffusion models (Sohl-Dickstein et al., 2015) have taken the world of image generation by storm, advancing the state of the art further than ever before (Rombach et al., 2021; Ramesh et al., 2022). Can the same framework encounter as much success on the text modality? Diffusion for language is indeed an attractive prospect. Compared to autoregressive (AR) models (Bengio et al., 2000; Sutskever et al., 2011; Austin et al., 2021; Hoffmann et al., 2022), diffusion models can predict all tokens in a sequence at once. This allows for bidirectional, rather than causal attention—increasing interactions between tokens, potentially leading to more coherent samples. Diffusion models can make a better usage of hardware accelerators during inference than AR models, since computations are parallelizable over the sequence axis. Yet AR models remain the mainstream approach for modelling text. A major obstacle to text diffusion is that diffusion processes typically operate in continuous space. While this naturally handle images, text is inherently discrete. Consequently, most previous attempts to apply diffusion to text have focused on *discrete* diffusion-like approaches. These methods do not benefit from the refinements made to continuous diffusion in the image domain. Crucially, they cannot make use of guidance (Dhariwal & Nichol, 2021), which drastically improves diffusion models sample quality.

We address this gap by making a simple observation: language models operate mostly in continuous space, with discrete tokens only as inputs and outputs. A natural idea is then to conduct diffusion directly in a continuous token embedding space. For simplicity, we use a fixed embedding space, either random or stemming from a trained language model. Combined with the "self-conditioning" (Chen et al., 2022) refinement, this forms the basis of the method we propose, Self-conditioned Embedding Diffusion (SED). SED models rival mainstream AR models in both conditional and unconditional text generation. We make the following contributions:

- In section 3, we introduce SED, the first continuous diffusion approach for text with good scaling properties (testing models up to 420M parameters). We analyze several continuous text diffusion settings, and identify self-conditioning and diffusion on small fixed embeddings as key factors to make continuous text diffusion work.
- In section 4, we apply classifier-free guidance (Ho & Salimans, 2022) to text data—an original achievement. We show that SED can rival AR models on generic language tasks, for similar models sizes. SED samples achieve a better likelihood-entropy trade-off compared to these models, and are deemed comparable (if slightly worse) by human raters.

## 2 RELATED WORK

We provide an overview of diffusion models with a focus on modeling discrete data, as well as AR models and sample-based metrics for evaluating text generation.

**Continuous diffusion on continuous image data.** Continuous diffusion has recently established itself as the method of choice for modeling continuous data such as images. While our main focus in this paper is on discrete data, we review some key works in continuous data modeling as this literature was the major source of inspiration for SED. The first continuous diffusion formulation was introduced in the seminal work by Sohl-Dickstein et al. (2015). Ho et al. (2020) improved and simplified this formulation, relating it to denoising score matching, and creating a new method called DDPM. Nichol & Dhariwal (2021) further improved upon DDPM, showcasing impressive diffusion results compared to GANs. Rombach et al. (2021, Stable Diffusion) introduced diffusion in latent space. Conceptually similar to SED, it was specifically targeted at image modeling. Classifier-free guidance was proposed by Ho & Salimans (2022) as a mean to improve image fidelity at the cost of reduced diversity. GLIDE (Nichol et al., 2022) scaled up the ideas of guided diffusion, while DALL-E 2 (Ramesh et al., 2022) and Imagen (Saharia et al., 2022) are the latest, most advanced image generation systems to date, combining most of the improvements proposed in previous works.

**Discrete diffusion on discrete data.** One cannot simply reuse the methods that are successful on continuous image data in the discrete text domain. A number of bespoke methods have been explored instead, forming the family of *discrete diffusion* approaches. In discrete diffusion, the data is corrupted by switching from one discrete value to another. This was first proposed in the seminal work by Sohl-Dickstein et al. (2015), where it was tested on simplistic binary heartbeat data. It was extended to multinomial text modeling (Hoogeboom et al., 2021) and further scaled up in the D3PM work (Austin et al., 2021). Most recently, a similar discrete diffusion approach was applied to image modeling in VQ-Diffusion (Gu et al., 2022). In parallel, a few diffusion-like approaches were proposed in the denoising autoencoders literature. CMLM (Ghazvininejad et al., 2019) tackled machine translation. SUNDAE (Savinov et al., 2022) was the first non-AR method to show strong results both in machine translation and unconditional text generation. MaskGIT (Chang et al., 2022) demonstrated excellent results in modeling VQ-discretized images. These approaches rely on training models to predict masked tokens from their context, and iterating this reconstruction step multiple times at sampling time. Despite those positive developments, the samples from discrete diffusion methods for text modeling remains less coherent than those produced by AR methods.

**Continuous diffusion on discrete data.** Fewer works try to tackle diffusion on discrete data from the same angle as SED – starting by turning the data into continuous representations before modeling it with continuous diffusion formulations. Mittal et al. (2021) used a VAE to generate such representations for discrete music modeling, with exciting results. Closest to SED, Diffusion-LM (Li et al., 2022) trains a token embedding together with the diffusion model itself. Diffusion-LM meets success on specific language applications, in low data regime and on constrained, very formatted textual data. Most recently, Analog Bits (Chen et al., 2022) introduced *self-conditioning*, closely related to step-unrolls in SUNDAE (Savinov et al., 2022), together with bit-level modeling to improve the generation of discretized images. While the qualitative results of those continuous methods on text modeling show promise, they have not been shown to scale to large realistic text datasets like C4 (Raffel et al., 2020) yet, or to compare with AR approaches on generic language tasks.

**Auto-regressive modelling on discrete data.** AR models remain the method of choice for modeling discrete data. In combination with neural networks, they were first explored by Bengio et al. (2000) and later combined with RNNs (Sutskever et al., 2011). Their breakthrough moment came with the advent of the Transformer architecture, introduced by Vaswani et al. (2017) for machine translation. Even more impressive results were shown with GPT-3 (Brown et al., 2020), which trained a large AR language model unconditionally, and used few-shot prompting to adapt it to new tasks. A few works later improved upon the results of GPT-3, including Hoffmann et al. (2022).

**Sample-based evaluation of text generative models.** There are traditionally two classes of metrics for generative modeling: likelihood-based and sample-based. While the likelihood-based way is mathematically appealing, its usefulness for measuring progress is reduced by the fact that not

all models readily provide likelihood computation. Just like the sampled-based FID metric was important for driving the progress of diffusion in image modeling, there is a need for a sample-based metric which would be universally accepted for text modeling. Caccia et al. (2018) investigated fidelity/variance metrics for evaluating text GANs. Semeniuta et al. (2018) suggested using FID for texts. De Masson d'Autume et al. (2019) later used those previously proposed metrics to iterate on ScratchGAN but did not provide conclusive guidance on which metric a practitioner should choose – essentially finding serious vulnerabilities in all investigated metrics. We opted for a middle ground, reporting both sample likelihood according to a strong AR model and human preferences.

## 3 METHOD

In this section, we outline the different components of SED: continuous diffusion in the space of token embeddings and self-conditioning, which form the basis of our approach for unconditional text generation; span masking and guided diffusion to enable conditional generation.

### 3.1 DIFFUSION MODELS FOR UNCONDITIONAL TEXT GENERATION

**Diffusion models in continuous space.** We consider diffusion models as introduced by Sohl-Dickstein et al. (2015) and improved by Ho et al. (2020). A diffusion model aims at modelling a data distribution $x_0 \in \mathbb{R}^n \sim q \in \mathcal{D}(\mathbb{R}^n)$ by estimating a sequence of latent variables $x_T, ..., x_1$ of the same dimensionality as the data $x_0$. Starting from $x_0$, the latent variables are generated with a Markov chain called the *forward process*: $x_t \sim q(\cdot | x_{t-1}, t)$. It is defined by gradually interpolating the iterate with Gaussian noise according to noise levels defined by a schedule $\beta_1, ..., \beta_T$:

$$x_t \sim q(\cdot \,|\, x_{t-1}, t) = \mathcal{N}(\sqrt{1 - \beta_t} x_{t-1}, \beta_t I). \tag{1}$$

This parametrization gives us a closed form to sample $x_t$ for any arbitrary $t \geq 1$, given $x_0$:

$$x_t = \sqrt{\alpha_t} x_{t-1} + \sqrt{1 - \alpha_t} \epsilon_t = \sqrt{\overline{\alpha}_t} x_0 + \sqrt{1 - \overline{\alpha}_t} \epsilon, \tag{2}$$

where $\alpha_t := 1 - \beta_t, \overline{\alpha}_t := \prod_{s=1}^{t} \alpha_s, \epsilon_t \sim \mathcal{N}(0, I)$ and $\epsilon \sim \mathcal{N}(0, I)$.

We define our generative model by approximately inverting the diffusion process of Eq. 1 to obtain a *reverse process*. The reverse process starts from $x_T \sim \mathcal{N}(0, I)$ and is defined as a Markov chain with learned Gaussian transitions (parameterized by $\theta$, the weights of a neural network): $x_{t-1} \sim p_\theta(\cdot | x_t) = \mathcal{N}(\mu_\theta(x_t, t), \sigma(t)^2 I)$. We train a neural network to predict an estimate $\hat{x}_0(x_t, t, \theta)$ of the data $x_0$ and approximate the reverse process by using the following parametrization, with learnable means but fixed variances, and a fixed schedule $\beta_1, ..., \beta_T$:

$$\mu_\theta(x_t, t) = \frac{\sqrt{\overline{\alpha}_{t-1}} \beta_t}{1 - \overline{\alpha}_t} \hat{x}_0(x_t, t, \theta) + \sigma(t)^2 = \frac{1 - \overline{\alpha}_{t-1}}{1 - \overline{\alpha}_t} \beta_t. \tag{3}$$

While there exists a tractable variational lower-bound (VLB) on $\log p_\theta(x_0)$, Ho et al. (2020) showed that better results are obtained by optimizing a simplified objective that re-weights the terms in the VLB. We follow this approach, which simplifies the loss to a sum of mean-squared errors between the ground truth data $x_0$ and its estimates $\hat{x}_0(x_t, t, \theta)$:

$$\mathcal{L}_{\text{diffusion}} = \mathbb{E}_{x_0 \sim q(x_0),\, t \sim \mathcal{U}(1,T)} \| x_0 - \hat{x}_0(x_t, t, \theta) \|^2. \tag{4}$$

Though this framework works out of the box on images, which are close to continuous, we cannot apply it directly to the discrete tokens of the text modality. To resolve this issue, we perform continuous diffusion in a continuous space in which we embed text tokens.

**Diffusion on word embeddings.** We consider textual data $w = (w_1, \ldots, w_N)$, where each $w_i$ is a one-hot representation in $\mathbb{R}^V$ of a discrete token in $\{1, ..., V\}$. Each token $w$ has an associated embedding $e_w \in \mathbb{R}^D$, with fixed norm $\sqrt{D}$ to match the norm of a random gaussian sample in dimension $D$ used to noise clean data. We denote by $E \in \mathbb{R}^{D \times V}$ the matrix of all embeddings.

We define our diffusion process in embedding space, rather than in token space. To that end, we define a forward *discrete-to-continuous* step $q_{\boldsymbol{V}}(\boldsymbol{x}_0|\boldsymbol{w}) = \mathcal{N}(\boldsymbol{E}\boldsymbol{w}, \sigma_0^2\boldsymbol{I})$, where $\sigma_0$ is a constant scale factor with a similar order of magnitude as $\beta_1$. Conversely, we define a reverse *continuous-to-discrete* step $p_{\boldsymbol{R}}(\boldsymbol{w}|\boldsymbol{x}_0) = \prod_{k=1}^N \mathrm{Cat}(w_k|\boldsymbol{E}'(\boldsymbol{x}_0)_k)$, where $\boldsymbol{R} \in \mathbb{R}^{V \times D}$ is a learnable readout matrix initialized to $\boldsymbol{E}^\top$ and $\mathrm{Cat}(w_k|\boldsymbol{l})$ is the softmax probability of token $k$ with logits $\boldsymbol{l} \in \mathbb{R}^V$.

To train the readout step, we add a reconstruction loss to $\mathcal{L}_{\mathrm{diffusion}}$ during training. Conveniently, it naturally arises when deriving the VLB of $p_\theta(\boldsymbol{w})$ with this discretization step (Li et al., 2022), introducing a simple cross-entropy loss to maximise $p_\theta(\boldsymbol{w}|\boldsymbol{x}_0)$:

$$\mathcal{L}_{\mathrm{recon}} = \mathbb{E}_{\boldsymbol{w} \sim \mathcal{D}, \boldsymbol{x}_0 \sim q_{\boldsymbol{V}}(\boldsymbol{w})}[-\log p_{\boldsymbol{R}}(\boldsymbol{w}|\boldsymbol{x}_0)], \qquad \text{with} \qquad \mathcal{L}_{\mathrm{total}} = \mathcal{L}_{\mathrm{diffusion}} + \mathcal{L}_{\mathrm{recon}}. \quad (5)$$

Contrary to what is done in Li et al. (2022), we do not learn the embedding matrix $\boldsymbol{E}$, as we identified that it was empirically unstable and could lead to drops in unigram entropy. The reconstruction loss $\mathcal{L}_{\mathrm{recon}}$ therefore only depends on the trainable readout weights $\boldsymbol{R}$.

At sampling time, we run the reverse process for $T = 1000$ steps, ultimately yielding a continuous embedding $\overline{\boldsymbol{x}}_0$ of size $d_{\mathrm{embed}}$. We multiply it by $\boldsymbol{R}$ to obtain logits in $\mathbb{R}^V$, and then use the index of the maximum component to convert it to a token $w_i$, with $i = \arg\max_{1 \le j \le V}(\boldsymbol{R}\overline{\boldsymbol{x}}_0)$. This entails running $T$ full forward passes which is quite expensive compared to cached AR sampling; however each forward pass computes all timesteps at once which is naturally parallelisable. Further, we hope to benefit from many diffusion sampling improvements to get $T$ down to low double-digits.

**Self-conditioning (Chen et al., 2022).** In standard diffusion sampling, at each timestep $t$ the denoising network generates an estimate $\tilde{\boldsymbol{x}}_0 = \hat{\boldsymbol{x}}_0(\boldsymbol{x}_t, t, \theta)$ of $\boldsymbol{x}_0$ given only $\boldsymbol{x}_t$ as input. Self-conditioning progressively refines $\boldsymbol{x}_0$ estimates by passing the estimate $\tilde{\boldsymbol{x}}_0^{t+1}$ obtained at the previous sampling step as input to the denoising network; the self-conditioned estimate is then defined as $\tilde{\boldsymbol{x}}_0^t = \hat{\boldsymbol{x}}_0^t(\boldsymbol{x}_t, \tilde{\boldsymbol{x}}_0^{t+1}, t, \theta)$, and sets the diffusion direction. In practice conditioning is performed by concatenating $\boldsymbol{x}_t$ and $\tilde{\boldsymbol{x}}_0^{t+1}$ on the feature axis. To approximate the inference behavior at train time while remaining computationally efficient, we compute a first estimate $\overline{\boldsymbol{x}}_0^t = \hat{\boldsymbol{x}}_0(\boldsymbol{x}_t, 0, t, \theta)$ with the self-conditioning set to zero, then perform a second forward pass using a stop gradient on $\overline{\boldsymbol{x}}_0^t$ to obtain $\tilde{\boldsymbol{x}}_0^t = \hat{\boldsymbol{x}}_0(\boldsymbol{x}_t, \overline{\boldsymbol{x}}_0^t, t, \theta)$. The denoising network is then optimized using the output from the two forward passes in order to estimate $\boldsymbol{x}_0$ accurately with and without self-conditioning.

Equipped with these 3 components we can train models to generate text, though only unconditionally. To add conditional generation to our system's capabilities, we use two additional methods.

## 3.2 SPAN MASKING AND GUIDANCE FOR CONDITIONAL TEXT GENERATION

By design diffusion models for text generation are flexible and can handle a wide variety of infilling tasks. This is a key advantage over the predominant auto-regressive language models that typically generate text in a left-to-right fashion.

**Span masking.** We train our model on a rich set of infilling tasks with the following method. We optimize the diffusion loss from Eq. 4 only over $\boldsymbol{x}$ while fixing the conditioning tokens $\boldsymbol{c}$. Conditioning tokens $\boldsymbol{c}$ are defined by a binary conditioning mask $\boldsymbol{m}$ set to one on conditioning positions and zero on positions to be infilled.

We sample conditioning mask $\boldsymbol{m}$ randomly as follows. Given a sequence of length $L$ and a maximum number of spans $M$, we sample a number of spans $n$ uniformly in $[1, M]$. Span starting positions are defined by $n - 1$ integers $(i_1, ..., i_{n-1})$ sampled uniformly without replacement and sorted in increasing order to satisfy $0 < i_1 < ... < i_{n-1} < L$. The tuple $(i_1, ..., i_{n-1})$ partitions the sequence of tokens in $n$ spans satisfying $\mathbb{E}[i_k|n] = \frac{k}{n}L$. The conditioning mask $\boldsymbol{m}$ is defined using even spans for conditioning and odd spans for infilling, and then $\boldsymbol{m}$ is flipped with a $50\%$ probability. The case $n = 1$ corresponds to unconditional generation; we then set $\boldsymbol{m}$ to 0 everywhere.

This span masking strategy defines a collection of text generation tasks with a large variety of conditioning which on average evenly splits the sequence between conditioning and infilling spans. It enables conditional generation, and opens the door for additional diffusion improvements.

**Guided diffusion.** Guidance (Dhariwal & Nichol, 2021) often improves the sample quality of conditional diffusion models. We use *classifier-free* guidance (Ho & Salimans, 2022), which alleviates the need for a separately-trained guide model. In the conditional case, our estimator $\tilde{\boldsymbol{x}}_0$ is now

Table 1: SED samples on unconditional generation, fill-in-the-middle and several spans in-filling.

| Task | Samples |
|------|---------|
| Unconditional | We make use of the very best supplies and solutions to ensure that the work is going to stand up to the test of time, and we help you save money with techniques that do not change the quality of your mission. We'll achieve this by offering you the best deals in the field and avoiding pricey mistakes. If you want to spend less, Refrigerator Unit Repair Guys is the company to contact. |
| Fill-in-the-middle | A year ago in Paris, I had the opportunity to take a field trip to La Rite-en-Laurences International de France where I met David Nigel Johnson, a professor of social studies. What a great trip and what a great day! |
| Spans in-filling | There was no evidence, only fleeting glimpses of the killer and his fate. In fact, it seemed that there was no evidence. It was all guesswork, and one of the most unusual murder cases throughout history. |

a function $\hat{\boldsymbol{x}}_0(\boldsymbol{x}_t, \boldsymbol{c}, \tilde{\boldsymbol{x}}_0^t, t, \theta)$, where $\boldsymbol{c}$ are fixed conditioning tokens. During training, with fixed probability the conditioning tokens $\boldsymbol{c}$ used in the estimator $\hat{\boldsymbol{x}}_0$ are dropped and set to a null label $\emptyset$. During sampling, the model prediction is extrapolated in the direction of $\hat{\boldsymbol{x}}_0(\boldsymbol{x}_t, \boldsymbol{c}, \tilde{\boldsymbol{x}}_0^{t+1}, t, \theta)$ and away from $\hat{\boldsymbol{x}}_0(\boldsymbol{x}_t, \emptyset, \emptyset, t, \theta)$ as follows:

$$\tilde{\boldsymbol{x}}_0^t = \hat{\boldsymbol{x}}_0(\boldsymbol{x}_t, \emptyset, \emptyset, t, \theta) + s \cdot \left( \hat{\boldsymbol{x}}_0(\boldsymbol{x}_t, \boldsymbol{c}, \tilde{\boldsymbol{x}}_0^{t+1}, t, \theta) - \hat{\boldsymbol{x}}_0(\boldsymbol{x}_t, \emptyset, \emptyset, t, \theta) \right), \quad (6)$$

where $s \geq 1$ is the guidance scale. Remark that we jointly drop conditioning *and* the self-conditioning $\tilde{\boldsymbol{x}}_0^{t+1}$, concretely setting both values to zero. Classifier-free guidance allows leveraging both the unconditional and conditional abilities of a model to improve its conditional generations.

## 4 EXPERIMENTS

### 4.1 TRAINING DETAILS

We train all our models on the C4 dataset (Raffel et al., 2020), using a SentencePiece tokenizer (Kudo & Richardson, 2018) composed of 32000 words. We use a non-causal transformer model (Vaswani et al., 2017) as our diffusion model (see Appendix A for details). SED models are trained with sequence length 256, while for ablations models are trained with sequence length 128. We insert uniformly, i.e. not necessarily at the end of the sequence, 10% of padding tokens in the training set to allow SED models to generate samples of varying size and provide more flexibility.

To generate word embeddings, we train a BERT model of fixed size (150m parameters) and feature dimension $d_{\text{model}} = 896$. The diffusion space is defined by the initial lookup table of this BERT model. We bottleneck the dimension of the word embeddings $d_{\text{embed}}$ and add a linear projection layer from $d_{\text{embed}}$ to $d_{\text{model}}$ at the beginning of the model. We found this helped diffusion (see section 4.4).

SED models are trained with a cosine noise schedule (Dhariwal & Nichol, 2021), with $\beta_1 = 2.10^{-3}$, $\sigma_0 = 10^{-2}$ and $T = 1000$. We use batches of 65.536 tokens, thus for sequence length 256 the batch size is set to 256. We use a maximum span count of 5 for all runs except for its specific ablation. We train SED models at two different scales: SED-S (135m parameters, $10^6$ training steps) and SED-L (420m, $2.10^6$ steps). Their detailed architectures can be found in Appendix A.

### 4.2 VALIDATION

While optimizing the perplexity of AR models for text leads to improved language models, directly optimizing the ELBO of diffusion models for images does not correlate strongly with sample quality as observed by Nichol & Dhariwal (2021); Kingma et al. (2021); Ho & Salimans (2022). For images, the sample based metric FID (Heusel et al., 2017) has been introduced as a measure of sample quality and is now widely adopted. Similarly, we need a sample-based metric for text generation that is reliable and allows comparison between a large variety of generative models. To provide a fair comparison to AR models, we rely on three metrics.

The first metric measures how likely the samples produced by a model are according to an AR language model with 70B parameters, trained on 1.4B tokens (Hoffmann et al., 2022); we denote

this metric AR NLL for auto-regressive negative log-likelihood. It provides a continuous measure of sample quality that has proven useful when combined with a measure of sample diversity, e.g. in the development of nucleus sampling (Holtzman et al., 2020) for improved AR model decoding.

To measure diversity we rely on a second metric, the unigram entropy of samples, which helps balance the AR NLL that can be gamed by unnatural repetitive samples. For both these metrics, our target is the score of the validation set data. Deviating from the data unigram entropy in particular is a sign of degenerate modeling.

Though this initial combination has provided us with a reliable signal to iterate over our model design, it remains imperfect; it too can be gamed, though it is harder to do so. To address this limitation, we also report human preferences. We presented 6 colleagues with 20 pairs of samples for each comparison, asking them to pick the best one. For all three metrics, we report results on two tasks: unconditional language modeling and suffix in-filling, the later a heavily conditioned task.

### 4.3 RESULTS

**Samples.** We present samples generated with our SED models in Table 10. We use a single model to perform a wide variety of text generation tasks, such as unconditional generation, filling-in-the-middle or filling several spans of text. We show strong performance in the unconditional case, with samples that are syntactically correct and stay coherent on long sequences. In the conditioned case, SED models are able to infill spans with coherent transitions and links to the conditioning but also exhibit a rich diversity. By design, SED yields flexible bi-directional masking models that can perform text generation on a diverse set of conditioned task. To compare SED with AR baselines we next restrict conditioning to a prefix and consider a task of suffix in-filling.

**Comparison to AR models.** To assess the generation ability of SED, we compare against AR baselines of similar capacity and trained following optimal scaling laws from Hoffmann et al. (2022) on suffix in-filling. We sample a batch of sequences from C4 and use the first 128 tokens as conditioning given to the model to generate a suffix of 128 tokens. Figure 1 reports AR NLL and unigram entropy of the generated suffixes for AR and SED models. As a

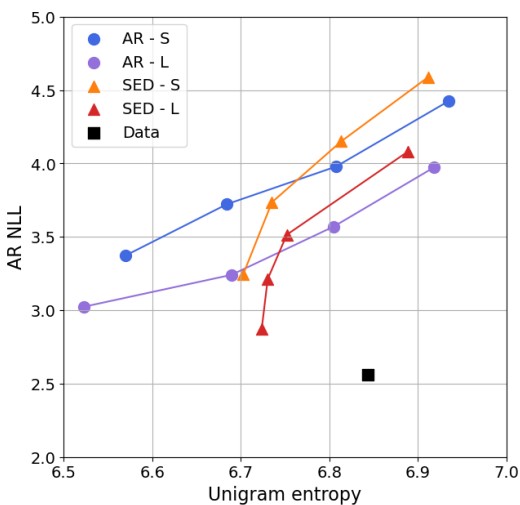

Figure 1: Comparison of sample quality and diversity of SED versus AR models on suffix in-filling. SED uses guidance with scales in $\{1, 2, 4, 8\}$ and AR uses nucleus sampling with a top-$p$ in $\{1.00, 0.95, 0.90, 0.85\}$. Top-right points are SED models with a guidance scale of 1 or AR models with a top-$p$ of 1.

reference point, we compute the AR NLL and unigram entropy of the ground truth C4 suffixes and report it on the plot. Several methods can be used to improve sampling quality at the cost of samples diversity; we use nucleus sampling (Holtzman et al., 2020) for AR models and guidance (Dhariwal & Nichol, 2021; Ho & Salimans, 2022) for SED models. We show the impact of guidance on samples quality in Table 3. To our knowledge, we are the first to show sample quality improvement when using guidance for text generation.

As shown in Figure 1, both SED-S and SED-L perform strongly when compared against AR baselines – even though we report a metric favoring AR models on a task AR models have been designed to optimize. Similar to nucleus sampling for AR models, guidance has a strong positive impact on sample quality that is both observed quantitatively with improved AR NLL in Figure 1 and qualitatively in Table 3. We observe that using a top-$p$ nucleus sampling below $0.8$ for AR models or a guidance scale above $4$ for SED models leads to samples exhibiting a lot of repetitions, a degenerate case reflected by a lower entropy of samples even though sample AR NLL improves.

Table 2: We compare SED-L (guidance scale 2.5) and AR-L (nucleus sampling, $p = 0.95$) samples.

| SED L | AR-L |
|---|---|
| You're going to love wearing this traditional tee from our latest Wilson collection. Designed in a scrapped floral styled knit with a sleeve of asymmetric lines across the round sole. Lightly fluffy, the square pleats will take you right to | A Koda Ram 25 is presented in sedan and a Maxima saloon. Based on the Acenta car, the powerful Koda 2014 hits Indian roads in the "Maxima" body-con shape. Being powered by a Hyundai i20 1.4 litre diesel engine, the Koda 2014 is coupled |
| The beaver is an interesting animal that lives in rivers and lakes. He is not mainly concerned with finding wolves and dolphin but also has a great hunger for fish. The beaver has sharp legs, large eyes, and a black coat | The beaver is an interesting animal that lives in rivers and streams. It is usually seen in big numbers in the fields or upstream, and is quite docile. On cold days when its pattern is perfect, the beaver will have some interesting, and sometimes |
| Once upon a time in Spain, Leonardo Puelva had the pleasure of meeting guests at Spanish restaurant, Buva Casinos. While driving, he got a chance to get to know the people behind the restaurant and, of course, how they made his experience very interesting. After his conversation, he got to | Once upon a time in Spain, which seems pretty much the same way now, the question that was posed to each of us at the end of our interview was "would you like to see Froome one day?" In retrospect, after our interview, we have grown ever closer to that answer. As you will read in the article, I know that |

Table 3: Impact of guidance on samples quality using our SED model.

| Guidance | 1.0 | 2.5 | 5.0 |
|---|---|---|---|
| | In the cold, cold night sky, a fairy princess sits in a chair and surrounded by tea leaves in a pond. Meanwhile, she bies back into the cold, with bluish hair on her hips and elbows on her forehead - and her fingers numbed by the freezing temperature. | In the cold, cold night of November 2018, a little girl sits in a chair hidden under a light blanket on a patio. Meanwhile, she bends back into the chair with bluish hair on her forehead, her hands on her face, her fingers numbed by the freezing temperature. | In the cold, cold night of December, my oldest daughter sits in a chair accentuated in cotton fabrics and a pillow. Meanwhile, she yearns straight in the cold air, her wrists covering her neck, her eyes straight on her forehead, and her fingers numbed by the freezing temperature. |
| | Barbara was one of our many wonderful women that really helped so I am so blown off by her purpose, civility; and adversity. Once I started interacting with her, it proved to me that no matter how hard this was, she always strove for excellence. | Barbara was one of the most gifted women in the world. She was creative and stood up by her integrity and civility; against adversity. Although she placed herself higher than her peers, it proved to me that no matter how hard this was, she always strove for excellence. | Barbara was one of the most brilliant women in the world. She was amazing in her heart, her spirit, her mind and in the soul. She never turned people off in her absence. It proved to me that no matter how hard this was, she always strove for excellence. |

Our human preference scores temper our observations in Table 4. They show that our NLL and entropy metrics do not tell the whole story, as humans still prefer AR models at equivalent size. While SED-L performs slightly worse than AR-L (38% preference in suffix in-filling, 44% on unconditional generation), its scores remain comparable. SED-L is roughly on par with AR-S.

Finally, we compare SED and AR models' qualitative examples with short prompts in Table 2.

Table 4: SED-L vs other models human preference scores on conditional and unconditional tasks.

| | SED-S (cond) | AR-S (cond) | AR-L (uncond) | AR-L (cond) |
|---|---|---|---|---|
| SED-L | 63.4% $\pm$ 4.3% | 51.0% $\pm$ 5.0% | 43.8% $\pm$ 4.4% | 37.7% $\pm$ 4.4% |

### 4.4 ABLATIONS

Table 5: Ablation of the proposed SED approach on unconditional generation. Both self-conditioning and embeddings pretraining play a key role in the model performance.

| Diffusion space | Self-conditioning | Unigram entropy | AR NLL |
|---|---|---|---|
| Bits (Chen et al., 2022) | ✗ | 6.97 | 7.01 |
| | ✓ | 7.47 | 6.05 |
| Random embeddings | ✗ | 6.90 | 6.80 |
| | ✓ | 6.86 | 5.31 |
| Pretrained embeddings | ✗ | 6.75 | 5.66 |
| | ✓ | 6.77 | **4.57** |
| Data | | $6.70 \pm 0.04$ | $1.81 \pm 0.15$ |

**Self-conditioning and embedding pretraining.** Results from Table 5 and samples from Table 6 show the influence of both the diffusion space and self-conditioning. AR NLL decreases very significantly when using self-conditioning, regardless of the rest of the setup. Diffusing at the bit-level (Chen et al., 2022) yields very high NLLs. While using random embeddings performs markedly better, using pretrained embeddings results in further improved numbers.

Samples from Table 6 highlight that models trained on random word embeddings exhibit topic modelling abilities with the co-occurrence of words like *child* and *mother* even though the paragraph remains globally incoherent and meaningless tokens like *gluc* are generated. Self-conditioning dramatically improves sample quality; the diffusion model gets the low-level structure right and generates syntactically correct sentences, even though the global text is not intelligible. Combining self-conditioning and pretrained embeddings leads to globally coherent paragraphs that stay on topic with proper sentence structure.

**Embedding dimension.** An important design choice for SED is the word embeddings space. We study the influence of pretrained embedding size in Table 7. Surprisingly, there is a threshold after which performance degrades when increasing the dimension of embeddings. We visualize the forward process for different embedding sizes by displaying the nearest neighbor of a noised token while running the forward process. In high dimension we observe that the nearest neighbor of a noised token remains the starting token itself until it switches to a completely random, unrelated token. In low dimension, we often observe that the closest neighbor of a noised token goes through several semantically related tokens (nearest neighbor of the starting token) before ultimately becoming random. We hypothesize that the random walks defined by diffusion are more likely to drift towards neighbors of the starting token in low dimension. As a result, when diffusing in low dimension information is destroyed in a more semantically meaningful fashion, which leads to an easier learning problem for the denoising function.

**Number of spans.** In order to enable in-filling, we train the model not only to do unconditional generation but also to conditionally fill spans of tokens. For each data point we sample a span number uniformly at random and span delimiters to generate the span mask. Picking the maximum allowable number of spans has a significant effect on model performance, as we can see in Table 8. Somewhat counter-intuitively, adding span masking improves even unconditional generation NLLs. It also appears that using a relatively high maximum span number is optimal. We hypothesize that this results in a varied mix of task difficulty at training time, between "easy", very conditioned problems on the one hand and "harder", unconditional ones on the other.

**Scaling.** We show encouraging results when scaling from SED-S (150m) to SED-L (420m). We train both models on sequences of 256 tokens and report a AR NLL of 4.20 for SED-S compared to 3.68 for SED-L. This improvement translates to improved sample quality, as is confirmed by our human preference scores, which are much higher for the larger model (63%, see Table 4).

## 5 LIMITATIONS

While our results are promising and show that continuous diffusion for text can be an exciting alternative to AR models, the current approach does present some significant limitations.

Table 6: On unconditional generation, self-conditioning results in better sentence modelling, pretrained embeddings enhances topic modelling.

| Random embeddings | Random embeddings with self-conditioning | Pretrained embeddings with self-conditioning |
|---|---|---|
| did the buildingroom granted a lighter distance. On it though, salaries about clients that a child, which dispersed gluc so many events and certainly wanted the Mother's project, discovered by their child would keep | Tree brings the sound, bearing features and capabilities that we are set in. For the first time, she uses a customizable framework to use that we help students publicly solve the weather conditions that we only offer students | almost six decades ago - 72 percent of Americans didn't feel they'd actually rent their own cars this year. Conversely, 90 percent of Americans feel that the decision to rent a car is something they feel it's impossible |

Table 7: Word embeddings with small dimension have higher AR likelihood.

| Embed. dim. | 16 | 32 | 64 | 128 | 256 | 896 |
|---|---|---|---|---|---|---|
| AR NLL | 4.65 | **4.57** | 4.71 | 4.61 | 4.77 | 4.92 |

Table 8: Span masking tasks improves unconditional text generation.

| Max span count | 1 | 3 | 5 | 7 | 9 | 11 |
|---|---|---|---|---|---|---|
| AR NLL | 4.99 | 4.82 | 4.82 | 4.67 | **4.48** | 4.56 |

First, much more could be done in terms of model tuning, including scaling to much bigger models to better understand SED's limits, and to be able to compare it with state-of-the-art AR models. Our training regime in particular would certainly benefit from more hyperparameter optimisation.

Second, one compelling reason we chose to explore continuous diffusion for text is to leverage the improvements produced by the literature on image generation. While we have ported some (e.g. self-conditioning), a lot more remains unexplored. The most obvious example is the sampling process itself, where the number of required steps has been considerably reduced for images (e.g. Karras et al. (2022) goes from 1000 to 35, and Salimans & Ho (2022) all the way down to 4 on simple images). Our current sampling is very inefficient, and this direction is one of the first improvements to make over SED.

Third, SED crucially relies on diffusing in a pretrained embedding space. This means relying on a second model, and using embeddings that may not be optimal for diffusion. Ideally, we'd train the full model end-to-end, which could yield even better results. While Li et al. (2022) found some success with this approach, it was in a specific setting at a small scale; in practice we found it difficult to avoid competition between the diffusion and reconstruction loss.

Finally, our work would benefit from improved metrics in the experimental section. Because the current state of the art involves AR models, the field lacks established benchmarks for tasks diffusion models are potentially better suited for, such as text in-filling. We opted for a reasonable mix, evaluating the negative log-likelihood of generated samples according to a very strong AR model as well as their token entropy and complementing it with a human evaluation. However, both NLL and unigram entropy are gameable (e.g. AR models assign very low NLL to repetitive snippets, and long enough repetitions can fool even entropy). Further, our NLL is inherently tied to its AR model and could thus be providing an unfair advantage to AR models. All told, we still found both metrics quite useful for measuring research progress, and our human evaluation confirmed our results. Moving forward, defining a clean in-filling benchmark would help produce even more convincing results.

## 6 CONCLUSION

We propose SED, the first generally-capable continuous diffusion model for text generation. SED models can perform both conditional and unconditional generation, and their performance rivals AR models while being more flexible in their use (e.g. enabling in-filling). We demonstrate their performance and study the impact of the main design choices.

Despite its limitations, this work lays the foundation for more exciting research. Promising directions include speeding up the sampling following the lessons learnt in the image domain, devising better embedding spaces for diffusion and investigating new in-filling capabilities.

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

## A   MODEL ARCHITECTURE

For both the AR and the SED models, we use the same transformer (Vaswani et al., 2017) architecture, which are similar to those described in (Hoffmann et al., 2022), with relative positional encoding as described in (Dai et al., 2019) in the attention blocks, and with a 4x expansion and a Gelu (Hendrycks & Gimpel, 2016) non-linearity in the feed-forward blocks. The architecture hyper-parameters are detailed in table 9. Noised word embeddings, $x \in \mathbb{R}^{N \times D}$, are first passed through a linear projection that operates on each embedding independently to get a projected embedding whose feature dimension matches the width of the transformer, $d_{\text{model}}$. At diffusion step $t$, we compute a time embedding as a sinusoidal position embedding (Vaswani et al., 2017) of size $d_{\text{model}}$, which is then passed into a $d_{\text{model}} \times d_{\text{model}}$ linear layer and added to the projected embedding. We add a linear output projection layer $E'$ which takes the output of the transformer $y \in \mathbb{R}^{N \times d_{\text{model}}}$ and projects each element $(y_i)_{1 \leq i \leq N}$ back to the same size as the word embeddings. When using self-conditionning, we modify the input to the model by concatenating $x$ and $\hat{x}_0$ along the feature axis before passing them to the input projection layer.

Table 9: Model hyper parameters.

| Model | number of layers | $d_{\text{model}}$ | number of heads | head size |
|---|---|---|---|---|
| S | 12 | 896 | 16 | 64 |
| L | 12 | 1536 | 16 | 128 |

## B   FORWARD DIFFUSION PROCESS VISUALIZATION

To support the discussion on word embeddings dimension from Section 4.4, we present a visualization of the forward diffusion process. Given starting tokens $x_0$, we project the noised tokens $x_t$ of the forward process at step $t$ to their nearest neighbor among word embeddings $E$ to obtain $w_t$. We then store the 128 nearest neighbors $\mathcal{N}(w_0)$ of starting tokens $w_0 = x_0$ and define the rank $r_t$ of $w_t$ at its index in $\mathcal{N}(w_0)$. We display $w_t$ and highlight it in green if $r_t$ is close to zero (meaning $w_t$ is a close neighbor of $w_0$) and in increasingly red colors otherwise. We present the first 16 nearest neighbors of $w_0$ in Figure 2 and provide an illustration of the color code used for highlighting. Figure 3 shows an instance of the forward diffusion process while diffusing on embeddings of with a high dimension of 896 and Figure 4 shows diffusion on embeddings with a lower dimension of 32.

We observe that in high dimension, the noised token's closest neighbour remains the original token up until the point where any token could be its closest neighbour. The diffusion random walk does not seem to pass through the neighbourhoods of semantically-related tokens. We hypothesize that the root cause of this issue is that the embedding space is mostly empty (with only 32000 points in $\mathbb{R}^{896}$, as $d_{\text{embed}} = 896$); and that embeddings are potentially concentrating in a lower-dimensional space.

In contrast, in lower dimension we see meaningfully-related tokens appear as the corruption progresses ('brown' becomes 'grey', 'quick' becomes 'swift', 'over' becomes 'underneath' etc). We believe this more gradual information destruction is beneficial for the diffusion model.

## C   ADDITIONAL SAMPLES

| | 1 | 2 | 3 | 4 | 5 | 6 | 7 | 8 | 9 |
|---|---|---|---|---|---|---|---|---|---|
| 00 | the | quick | brown | fox | jumps | over | the | lazy | dog |
| 01 | The | fast | gray | wolf | swings | around | The | hungry | puppy |
| 02 | the | swift | grey | rabbit | moves | under | their | tired | pet |
| 03 | The | speedy | pink | spider | kicks | on | a | boring | horse |
| 04 | their | easy | yellow | deer | spins | about | our | silly | pup |
| 05 | his | simple | orange | lion | rides | through | the | lonely | dogs |
| 06 | their | slow | black | bull | hikes | in | its | bored | cat |
| 07 | your | quicker | purple | tiger | blows | after | your | impatient | animal |
| 08 | those | straightforward | red | frog | pulls | within | an | weary | Dog |
| 09 | these | faster | blue | superhero | knots | for | his | dumb | pig |
| 10 | its | rapid | straw | shark | loops | across | those | noisy | breed |
| 11 | my | instant | dark | bird | curves | off | these | crazy | pony |
| 12 | put | nice | coloured | crow | catches | up | this | stupid | goat |
| 13 | our | brief | redd | fur | pushes | out | my | anxious | kid |
| 14 | this | short | white | duck | draws | behind | The | awkward | cow |
| 15 | Our | prompt | cream | elephant | gears | against | any | miserable | monkey |

Figure 2: Nearest neighbors of tokens from the sentence "the quick brown fox jumps over the lazy dog"

| | 1 | 2 | 3 | 4 | 5 | 6 | 7 | 8 | 9 |
|---|---|---|---|---|---|---|---|---|---|
| 0 | the | quick | brown | fox | jumps | over | the | lazy | dog |
| 25 | the | quick | brown | fox | jumps | over | the | lazy | dog |
| 50 | the | quick | brown | fox | jumps | over | the | lazy | dog |
| 75 | the | quick | brown | fox | jumps | over | the | lazy | dog |
| 100 | the | quick | brown | fox | jumps | over | the | lazy | dog |
| 125 | the | quick | brown | fox | jumps | over | the | lazy | dog |
| 150 | the | quick | brown | fox | jumps | over | the | lazy | dog |
| 175 | the | quick | brown | fox | jumps | over | treatment | lazy | dog |
| 200 | thel | quick | brown | fox | flights | over | army | lazy | ole |
| 225 | stated | quick | brown | 25- | flights | over | army | lazy | Rams |
| 250 | makers | quick | tan | 25- | flights | over | army | Philip | Rams |
| 275 | furt | quick | tan | herical | job | vol | third | Rays | ball |
| 300 | Search | quick | abdomen | char | job | vol | third | pathway | thal |

Figure 3: Visualization of the forward diffusion process up to 300 steps when diffusing on embeddings with dimension 896.

| | 1 | 2 | 3 | 4 | 5 | 6 | 7 | 8 | 9 |
|---|---|---|---|---|---|---|---|---|---|
| 0 | the | quick | brown | fox | jumps | over | the | lazy | dog |
| 25 | the | quick | brown | fox | jumps | over | the | lazy | dog |
| 50 | the | quick | brown | fox | jumps | over | the | lazy | dog |
| 75 | the | quick | brown | fox | jumps | over | the | lazy | dog |
| 100 | the | quick | grey | fox | jumps | over | the | lazy | dog |
| 125 | the | Pinterest | gray | abduct | jumps | underneath | changed | lazy | animal |
| 150 | vivo | swift | gray | fox | hikes | over | No | bingo | dog |
| 175 | rapid | eager | grey | frog | umbled | over | ired | usi | collision |
| 200 | rapid | ggy | boy | mosqu | uttering | underneath | proud | casters | diesel |
| 225 | inter | ggy | blonde | mosqu | enced | underneath | Nor | gambling | tow |
| 250 | inter | tend | pink | illeg | acious | elsewhere | Yo | casino | cargo |
| 275 | un | !" | gray | erectile | dreaming | here | Nor | casino | bay |
| 300 | AI | !" | gray | thir | dreaming | here | Nor | � | rare |

Figure 4: Visualization of the forward diffusion process up to 300 steps when diffusing on embeddings with dimension 32.

**Reverse process**

```
1000  The be aver is an interesting animal that lives in rivers and decides 25,     143        ouses      Wednesday 2-   $4      7)       13.        2,000       (1980    guitarist   .4        none        1975.  5.
975   The be aver is an interesting animal that lives in rivers and 115           20.        2.         1991           1989,   great    thirty      14,         Thursday $1,000      Tuesday     specially   Friday      seventh     Mom    several
950   The be aver is an interesting animal that lives in rivers and toward 7%     1,000      21         14.            4-      2009,    57          1987,       Indiana  finally      $5          50          fabulous    Cell   enables
925   The be aver is an interesting animal that lives in rivers and toward takes  to         Wednesday  1989.          ...     honored  57          2,          until    2.3          163         Many        93          1970   Many
900   The be aver is an interesting animal that lives in rivers and 91            2007,      -10        Wednesday 2010). P      9),     November 2006, yesterday  fourth       dozens      December    $10.        2%     77
875   The be aver is an interesting animal that lives in rivers and their         2007,      90%        1992.          northern remarkable 9). November ambia  since    fourth       wonderful 40,  traveled   17-    UN
850   The be aver is an interesting animal that lives in rivers and traveled 2007, 90        16.        1997.          2010,   47      larger      18,         since    realized     wonderful [14  traveled   2%     UN
825   The be aver is an interesting animal that lives in rivers and 2013          1990.      16.        1997.          2010,   47      because     18,         successfully 10%     wonderful $10. traveled    13     8).
800   The be aver is an interesting animal that lives in rivers and toward (10     2010      16.        $4.            15,     2010.   because     many        2,       10%          wonderful moreover traveled  3.     among
775   The be aver is an interesting animal that lives in rivers and osl (10        2010      3%         Nov            needs   attractive larger  2,          2,       2009         20,000      5000        traveled    7%     among
750   The be aver is an interesting animal that lives in rivers and traveled (10   1982      3%         40             needs   !!!      2015        sites       completely $400      20,000      facebook    28.         7%     among
725   The be aver is an interesting animal that lives in rivers and 20-           5%         really     3%             135     needs   towards larger 118      1998,    1989         create      facebook    Washington 95      191
700   The be aver is an interesting animal that lives in rivers and 20-           (10        really     3%             95      needs   towards larger 118      2%       1989         create      facebook    1,200       and    58
675   The be aver is an interesting animal that lives in rivers and 2018          (10        1982       3%             95      needs   75,         larger      medieval 2,           millions    1987.       Wendy       1,200  and   58
650   The be aver is an interesting animal that lives in rivers and 51            (1980      17         3%             95      needs   75,         1997.       October  2,           Democrats extremely uckland  -40      135    58
625   The be aver is an interesting animal that lives in rivers and 1990,         13,        1982       31             $75     bathroom 75,        1997.       historic 2,           learned     extremely unfortunately 1981. 135  6)
600   The be aver is an interesting animal that lives in rivers and 1990,         2000.      literally 3%             38      needs   75,         1997.       historic 2,           because     1987.       taxi        Port   135   arrows
575   The be aver is an interesting animal that lives in rivers and 1990,         realize    3          continues 95    needs   towards 1997,      historic    2,       learned      1987.       company     Port        and    sights
550   The be aver is an interesting animal that lives in rivers and 300           realize    3          Pers           additionally needs 75,     27          waterfront rarely    learned      1987.       company     Harbor and   looks
525   The be aver is an interesting animal that lives in rivers and schools realize 3        continues  additionally needs 75,    27      waterfront  rarely      85       15,          museum      Port        and    looks
500   The be aver is an interesting animal that lives in rivers and schools realize 3        Pers       additionally relatively 75,  27      waterfront  rarely      85       1991,        2006        Port        and    --
475   The be aver is an interesting animal that lives in rivers and schools 6     3          ates       primarily 6.   75,     1890    central     rarely      those    1991,        2006        1981.       and    looks
450   The be aver is an interesting animal that lives in rivers and 55            usually    3          ates           primarily -28   1890   -28         rarely      28       While        1989,       Ocean       and    traveled
425   The be aver is an interesting animal that lives in rivers and roads         often      literally Pers            seventh -28     $300    27          coastal     rarely   among        1991,       1989,       Ocean  March looks
400   The be aver is an interesting animal that lives in rivers and lakes         .          literally ates            seventh -28     $300    modern      coastal     rarely   among        1991,       1986.       Ocean  March travels
375   The be aver is an interesting animal that lives in rivers and lakes         amazing literally ates               primarily southern 25,  1947        coastal     rarely   travelers    1991,       1986.       Ocean  and  traveling
350   The be aver is an interesting animal that lives in rivers and lakes         2015.      certainly ates            seventh -28     25,     modern      coastal     rarely   1990.        7,          1986.       Ocean  March ventures
325   The be aver is an interesting animal that lives in rivers and lakes         often      certainly ates            seventh southern 25.   1947        coastal     primarily among       1991,       1986.       Ocean  and  ventures
300   The be aver is an interesting animal that lives in rivers and lakes         .          certainly ates            seventh southern 25,   modern      coastal     works    10%          incredible Minnesota Ocean  and  traveled
275   The be aver is an interesting animal that lives in rivers and lakes         often      certainly ates            primarily southern 25,  modern     coastal     rarely   because      incredible Pacific   Ocean  and  travels
250   The be aver is an interesting animal that lives in rivers and lakes         often      certainly resides         primarily southern $300 western    picturesque out 10%    incredible Pacific   Ocean       and    travels
225   The be aver is an interesting animal that lives in rivers and lakes         2015.      certainly ates            primarily central 25,  western    metropolitan out 10%   incredible Pacific   Ocean       and    travels
200   The be aver is an interesting animal that lives in rivers and lakes         2015.      certainly floods          primarily central provided western northern disappeared 10% 70      Pacific     Ocean       and    travels
175   The be aver is an interesting animal that lives in rivers and lakes         2015.      It         ates           primarily central and  western     coastal     parts    10%          70          Global      Ocean  and  travels
150   The be aver is an interesting animal that lives in rivers and lakes         .          It         ates           primarily central and  western     coastal     rarely   10%          7:          Pacific     Ocean  and  travels
125   The be aver is an interesting animal that lives in rivers and lakes         .          It         resides        primarily central and  western     coastal     parts    10%          incredible Pacific   Ocean  and  travels
100   The be aver is an interesting animal that lives in rivers and lakes         .          It         resides        in        central and  western     coastal     parts    of           the         Pacific     Ocean  and  travels
75    The be aver is an interesting animal that lives in rivers and lakes         .          It         resides        in        central and  western     coastal     parts    of           the         Pacific     Ocean  and  travels
50    The be aver is an interesting animal that lives in rivers and lakes         .          It         resides        in        central and  western     coastal     parts    of           the         Pacific     Ocean  and  travels
25    The be aver is an interesting animal that lives in rivers and lakes         .          It         resides        in        central and  western     coastal     parts    of           the         Pacific     Ocean  and  travels
```

Figure 5: Reverse diffusion process of SED-L with guidance scale 2.5.

Table 10: Samples from SED-L.

| |
|---|
| This course is essential for the environment in which students choose the curriculum option study at JHD. Geographical Integration is at the heart of this department. Along with the urban infrastructural innovations of the mid-1990's and social issues in the 21st Century. People within the department regularly exemplify the concept of real integration. |
| That did trigger some rewarding words or insights to begin preparing their kid for their future. Luckily, Mikaela has been nice enough to tolerate my questions about what parents can do to help, even during her summer vacation. |
| I went to college at Boston University. After getting my degree, I decided to make a change! I enrolled in outdoor schools. After getting my degree, I loved jet skiing, offshore fishing, and Kitesurfing. The list became growing. More importantly, I started a career by fishing at sea. Now, I can't get enough of the Pacific Ocean! |
| The beaver is an interesting animal that lives in rivers and waterfalls across Puerto Rico. It loves kayaking, fishing, and swimming. But, you want to know what are the animals behind the beaver? |
| A year ago in Paris, I had the opportunity to take a field trip to La Rite-en-Laurences International de France where I met David Nigel Johnson, a professor of social studies. What a great trip and what a great day! |
| A year ago in Paris, my friends and I went on a dirt road trip to the city. I remember walking through the church, its beautiful square, narrow streets lit with memorials and passing a terrible Catholic Bishop I am used to - what a sad day... |
| There was no evidence, only fleeting glimpses of the existence of cognitive disability. There was no luck and no solid science. It was all guesswork, the blinding prospect of pneumonia could spur imagination at the possibility of brain damage. |

