# OpenReview forum: "Self-conditioned Embedding Diffusion for Text Generation"
_ICLR.cc/2023/Conference — Submitted to ICLR 2023_

### Official Review · Reviewer_gGpy · 2022-10-24

**Confidence:** 5
**Correctness:** 4
**Technical Novelty And Significance:** 3
**Empirical Novelty And Significance:** 2
**Recommendation:** 5

**Clarity, Quality, Novelty And Reproducibility:**

This paper is clearly written and easy to follow. The novelty of this paper lies in its adaptation of cutting edge techniques (classifier-free guidance, self-conditioning) at scale (C4 dataset; 420M parameter model) to the text diffusion setting. While I believe all the necessary information for reproducing these results is provided, there are a lot of details and reproducibility would be greatly improved by releasing code and the pre-trained models: I encourage the authors to do so.

**Strength And Weaknesses:**

This is a timely result on diffusion modeling for text generation, applying recent innovations in diffusion modeling (self-conditioning, classifier-free guidance) to text-domain diffusion modeling at a larger scale than previous work. Demonstrating the effectiveness of these methods on text data at scale is a valuable contribution.

That said, the evaluation of the quality of these models is somewhat lacking. Evaluation of open-ended generated text is a difficult and active area of research, which this paper largely ignores: for example (non-exhaustive list): (various metrics) Holtzman et al, 2019, (HUSE) Hashimoto et al., 2019, (Mauve) Pillutla et al., 2021. Instead of following guidance from the literature on evaluation, this work uses the following metrics: sample perplexity, unigram entropy (not a standard diversity metric), and head-to-head human preference using a small sample with limited nuance: "We presented 6 colleagues with 20 pairs of samples for each comparison, asking them to pick the best one." It is difficult to feel confident in the quality of these models based on these statistics and the limited samples presented in the text.

That said: here is what I took away from the evaluation. The human evaluation results (Table 4) do give me confidence that the SED models are somewhat reasonable: the SED-L (420M param) model is competitive qualitatively with the AR-S model. It is interesting (Table 5) that Analog Bits (Chen et al., 2022) performs poorly compared to the tailored solution for text (the performance differences are large enough that I believe them, despite concerns about the choice of metrics). On the other hand, I do not find the more subtle perplexity differences presented in Tables 7 and 8 convincing.

One question I have is: do the authors plan to release these models to the community? Releasing pre-trained text diffusion models of this scale would in itself would be a contribution, and to some extent mitigates concerns about the evaluation because members of the community would be able to use the model and reach their own conclusions.

**Summary Of The Paper:**

This paper describes an application of continuous diffusion models (Sohl- Dickstein et al., 2015; Ho et al., 2020 to text. Methodologically, this work differs from previous continuous text diffusion (Li et al., 2022) by leveraging recent insights on self-conditioning (Chen et al., 2022) and classifier-free guidance (Dhariwal & Nichol, 2021). Empirically, this work advances the state of the art for text diffusion by scaling these methods to a larger dataset (C4; Raffel et al., 2020) with larger models (135M and 420M parameter models) than considered by Li et al., 2022.

**Summary Of The Review:**

An interesting application of recent innovations in diffusion models to the text domain at larger scale than earlier work. The empirical evaluation of the resulting model is somewhat lacking.

---

### Official Review · Reviewer_Sxgy · 2022-10-24

**Confidence:** 4
**Correctness:** 4
**Technical Novelty And Significance:** 2
**Empirical Novelty And Significance:** 2
**Recommendation:** 5

**Clarity, Quality, Novelty And Reproducibility:**

The method in the paper is presented clearly, however, the novelty is limited. The results should be replicated without difficulties.

**Strength And Weaknesses:**

Strength:
+ This is one of the first continuous DLM works, and the paper explores the effectiveness of several key ideas in image diffusion in the text domain, namely the diffusion guidance and the self-conditioning.
+ The performance of DLM, even with minimal changes compared to its vision counterparts, is shown to be encouraging, about equal to or at worst, slightly worse than AR models.
+ Compared to Li et al. 2022, the authors tested the ideas on a larger scale data.

Weaknesses:
Most of the concerns I have are similar to what the authors have acknowledged in their limitation section.
+ The diffusion process requires a pretrained embedding space while having a similar results compared to AR models. This make the diffusion LM substantially more expensive and inefficient.
+ The data/model size are not large-scale
+ Most of the techniques used in the paper have been proposed perviously.

The author mentioned that "in practice we found it difficult to avoid competition between the diffusion and reconstruction loss". I would like to see more analysis on the difference between training the model from scratch and from the pretrained embedding space since Li et al. did start the diffusion without pretrained embedding. Does the authors find the training not convergent or the final model quality worse?

**Summary Of The Paper:**

The paper explores the Diffusion method for Text generation similar to the Diffusion Language Model (DLM, Li et al. 2022). Both this work and DLM propose to use continuous diffusion on the word embedding space to leverage recent advances in image diffusion techniques. This paper goes one step further and add self-condition (Chen et al. 2022) to the diffusion process to improve generation quality. This self-condition idea added the previous predicted embedding output of the previous diffusion step as a condition (through input concatenation) to the next diffusion step. The paper also proposes to use binary span masking to train the diffusion model in an infilling task and further explore diffusion guidance, as proposed in previous works for image diffusion.

As one of the first few works on diffusion language model, the results compared to SOTA AR models are mixed. Although there are some improvements in terms of token diversity and log-likelihood, the model still perform slightly worse in human evaluation.

**Summary Of The Review:**

The paper explores several recent techniques in image diffusions in text domain. The results are encouraging, but there are two key concerns with the paper. The first is the lack of novelty, the continuous diffusion has been explored in Li et al. 2022, and all the model improvements have been proposed in the vision domain. The second is the lower performance of DLM compared to SOTA auto regressive LM, even when trained on a pretrained embedding space. Having said that, the paper does show some interesting observation that can be a good foundation for future research.

---

> ### Public Comment · ~Carmen_Amo_Alonso1 · 2022-11-17
> **Reviewer's evaluation on " The results should be replicated without difficulties." is evidently wrong.**
>
> The reviewer's assessment that "The results should be replicated without difficulties." is evidently wrong:
>
> 1) The paper evaluation is done based on private models of DeepMind.
>
> 2) The word embedding used is not the one available from BERT and they have pretrained BERT on c4, which is also not available to public and is costly to pretrain.
>
> 3)  Most importantly, the paper has a sheer contrast with the prior work of Li et al, showing that pre-trained word embeddings does not work which is not discussed in the paper.
>
> 4) The paper has not released pre-trained model/code nor a promise to release them later. DeepMind also most probably do not have any plan to release Chinchilla, like many of their prior work.
>
> All in all, results are definitely not reproducible.

---

### Official Review · Reviewer_U4Mu · 2022-10-24

**Confidence:** 4
**Correctness:** 4
**Technical Novelty And Significance:** 2
**Empirical Novelty And Significance:** 4
**Recommendation:** 5

**Clarity, Quality, Novelty And Reproducibility:**

The work is presented clearly and is well written.
The authors make use of re-trained/private models to obtain word embeddings and to compute perplexity for evaluation. Using public checkpoints will aid in improving reproducibility and will allow for easier comparison with other work. Unless there is a reason to use private models, I would ask the authors to consider using released public checkpoints of language models.

**Strength And Weaknesses:**

Strengths
* The authors were able to scale diffusion models to work well when trained on large NLP datasets by using pretrained embeddings, self-conditioning (originally proposed in the Analog Bits paper) and a bottle neck for embedding dimension.
* The proposed method SED-L achieved lower perplexity and better unigram entropy when compared to an autoregressive model of similar capacity.
* The authors identified that classifier-free guidance can improve language generation results (in both the condition and unconditional setting).

Weaknesses
* It's not clear why a separate BERT model is trained from scratch, instead of using a publicly released checkpoint. Was it trained differently or on a specific dataset?
* The authors mention that using randomly initialized embeddings lead to instabilities in training and therefore propose using pertained embeddings. An ablation that would be of interest is to initialize the word embeddings with pre-trained embeddings, but also further finetune the embeddings as the model is trained.
* There are qualitative samples for different guidance values but it is hard to judge which is better and in what way. Presenting a quantitative evaluation with the different guidance value would be useful. Furthermore, the guidance explanation is missing some details. For instance, how are the conditioning tokens c provided to the model? Is x_t modified such that the tokens corresponding to the conditioning tokens are replaced with the ground truth?
* For bottlenecking, it is unclear whether bottleneck means project down from initial embedding dimension and back up to d_model or reduce the size of d_model.
* MAUVE (https://openreview.net/pdf?id=Tqx7nJp7PR) can be used as an additional evaluation metric to compare autoregressive models and SED models to increase confidence in the results.

**Summary Of The Paper:**

The authors present a continuous diffusion model that achieves performance comparable to autoregressive language models for natural language generation. The authors identify that self-conditioning and small fixed embeddings are important for achieving good performance with diffusion models for text generation.

**Summary Of The Review:**

The authors successfully show how diffusion models can be used with large language datasets. However, the authors mostly use techniques that are introduced in previous papers (self-conditioning, guidance). The most important contribution of this paper is to use pre-trained embeddings for diffusion. The paper can also benefit from additional results on other NLP tasks like controllable text generation.

---

### Official Review · Reviewer_avZf · 2022-11-01

**Confidence:** 3
**Correctness:** 3
**Technical Novelty And Significance:** 3
**Empirical Novelty And Significance:** 3
**Recommendation:** 6

**Clarity, Quality, Novelty And Reproducibility:**

This paper is clearly written and easy to follow. The ideas are mostly novel and seem reproducible.

**Details Of Ethics Concerns:**

As with any text generation models, there are concerns that this model might generate toxic, offensive and discriminatory text. Hence, this paper should be accompanied by an ethics statement which it currently does not seem to have.

**Strength And Weaknesses:**

Strengths:
1. This is the first work on diffusion LMs I have seen that show results on a simple unconstrained text generation baseline and seems to have positive results in terms of the metrics used.
2. The writing is clear and the motivations seem sound. There are some clever ideas incorporated like padding in the middle of the sequence, self-conditioning which seem to give the results a boost.

Weaknesses:
1. There is a discussion in limitations on the inadequacy of automatic metrics but the authors do not evaluate a variety of them that prior work has done such as MAUVE, distinct-n, zipf score, repetition rate and many more. These metrics should be computed to a get better idea of the quality and diversity of the output. Unigram entropy is not what is usually reported but would provide a similar trend to dist-1 I would imagine. Please see these this paper for the description of these metrics: https://arxiv.org/abs/2202.00666.
2. There are missing baselines such as Li et al 2022's diffusion LM which the authors describe as not being good for large datasets but this claim is not verified. The ablation in Table 5 with random embeddings is not the same loss function as far as I can tell as the diffusion LM paper.
3. The model rely on a pretrained BERT token embeddings which just seems like an arbitrary choice. An ablation should be provided with different pretrained embeddings to understand the impact of this choice.

**Summary Of The Paper:**

This paper presents a method to train text based diffusion models which can generate text non-autoregressively modeling bidirectional context. Since diffusion models work in continuous domains only, the authors propose the following changes to the standard setup (1) represent a sequence of tokens as a sequence of pretrained embeddings (obtained from a BERT-based model) and compute a standard diffusion training loss (L2), (2) to recover the discrete tokens from predicted embeddings, add another loss to the diffusion training loss, which computes the cross entropy wrt the target token by projecting the predicted vector back to the vocabulary space using a learnable matrix, and (3) add self-conditioning to this framework following Chen at al 2022. Trained a large corpus (C4), the generated LM shows improvement over autoregressive LM on the computed quality and diversity metrics. Ablations have been conducted on the proposed changes and show the importance of each setup.

**Summary Of The Review:**

The paper presents a diffusion language model by applying diffusion on pretrained embeddings. The evaluation seems to suggest that the outputs from this model have better quality and diversity scores than autoregressive LMs, however some important metrics are missing so it's difficult to clearly say if the model is better than the baselines.

---

### Public Comment · ~Carmen_Amo_Alonso1 · 2022-11-17
**This paper is not anonymous and the results are dubious and not reproducible**

Hi
I have major concerns about this paper:

1) **this is NOT an anonymous submission**, i.e., clearly this is from DeepMind since Chinchila model used for evaluation is not released to public. This breaks the anonymity requirement of submission to ICLR and does positively bias the reviewer's scores for this paper, which is unfair to other submissions.

2) The results of the paper are not reproducible, and the authors have not promised to release the codes either. The community therefore cannot benefit from this work. This paper follows the same model+objective as Li et al (https://arxiv.org/abs/2205.14217), except for additional self-conditioning on top of that for improvement over the baseline. However, as shown in the work of Li et al, usage of pretrained embeddings does not allow diffusion models train and Li et al emphasize to train the word embedding from scratch. So the baseline used as expressed in the paper should not work. Given that the other paper has a public code to test to certify it, this makes the results of this work dubious.

3) the method must be very sensitive on the pretrained embeddings as otherwise, the authors could have used the original pre-trained BERT embeddings rather than pretraining BERT from scratch on C4 dataset, which is rather a costly training procedure. This is not discussed in the paper.

4) The NLL loss metric used to evaluate the model is not necessarily correlated with the text quality. Prior works have shown that low perplexity of generated text is not necessarily an indication of high quality but of degenerate behavior (Nadeem et al., 2020; Zhang et al.,2021) and have proposed closeness to the perplexity of human-written text as a better evaluation. Reporting the human evaluation on very few examples, as done in the paper, is questionable, and does not provide a proper assessment for the quality of generated texts.

References:
1) Nadeem et al, A systematic characterization of sampling algorithms for open-ended language generation, AACL, 2020
2) Hugh Zhang, Daniel Duckworth, Daphne Ippolito, and Arvind Neelakantan. Trading off diversity and quality in natural language generation. In Proceedings of the Workshop on Human Evaluation of NLP Systems (HumEval), 2021.

---

### Decision · Program_Chairs · 2023-01-20

**Decision:**

Reject

**Justification For Why Not Higher Score:**

The paper of its current form is not ready. It would benefit from a careful revision to dismiss the concerns on reproducibility and fair and trustworthy evaluations.

**Justification For Why Not Lower Score:**

N/A

**Metareview: Summary, Strengths And Weaknesses:**

The main contribution of the paper lies in scaling and deploying several existing techniques for diffusion models, which have been shown to be successful for vision tasks, to text generation. While scaling existing techniques and applying to a new domain could be considered as a sufficient contribution and some of the empirical observations could provide valuable references for future investigations, there is a shared concern among the reviewers on the rigor and trustworthiness of performance evaluation. As both the code and pretrained models are neither available at this moment nor promised to be released to the public in the future, there is also concern on reproducibility and the diminished values to the community.